# Taxonomy, Phylogeny, Divergence Time Estimation, and Biogeography of the Family *Pseudoplagiostomataceae* (*Ascomycota*, *Diaporthales*)

**DOI:** 10.3390/jof9010082

**Published:** 2023-01-05

**Authors:** Zhaoxue Zhang, Xinye Liu, Mengfei Tao, Xiaoyong Liu, Jiwen Xia, Xiuguo Zhang, Zhe Meng

**Affiliations:** 1College of Life Sciences, Shandong Normal University, Jinan 250358, China; 2Shandong Provincial Key Laboratory for Biology of Vegetable Diseases and Insect Pests, College of Plant Protection, Shandong Agricultural University, Taian 271018, China

**Keywords:** *Pseudoplagiostomataceae*, biogeography, divergence times, morphology, new species, phylogeny

## Abstract

Species of *Pseudoplagiostomataceae* were mainly introduced as endophytes, plant pathogens, or saprobes from various hosts. Based on multi-locus phylogenies from the internal transcribed spacers (ITS), the large subunit of nuclear ribosomal RNA gene (LSU), partial DNA-directed RNA polymerase II subunit two gene (*rpb2*), the partial translation elongation factor 1-alpha gene (*tef1α*), and the partial beta-tubulin gene (*tub2*), in conjunction with morphological characteristics, we describe three new species, viz. *Pseudoplagiostoma alsophilae* sp. nov., *P. bambusae* sp. nov., and *P. machili* sp. nov. Molecular clock analyses on the divergence times of *Pseudoplagiostomataceae* indicated that the conjoint ancestor of *Pseudoplagiostomataceae* and *Apoharknessiaceae* occurred in the Cretaceous period. and had a mean stem age of 104.1 Mya (95% HPD of 86.0–129.0 Mya, 1.0 PP), and most species emerged in the Paleogene and Neogene period. Historical biogeography was reconstructed for *Pseudoplagiostomataceae* by the RASP software with a S–DEC model, and suggested that Asia, specifically Southeast Asia, was probably the ancestral area.

## 1. Introduction

*Pseudoplagiostomataceae* Cheew., M.J. Wingf. and Crous, a monotypic family, was introduced by Cheewangkoon, M.J. Wingf. and Crous, and *Pseudoplagiostoma* Cheew., M.J. Wingf. and Crous (type species: *Pseudoplagiostoma eucalypti* Cheew., M.J. Wingf., and Crous) was designated as the type of genus [1]. At present, *Pseudoplagiostoma* comprises ten species including *P. castaneae* T.C. Mu, J.W. Xia, and X.G. Zhang, *P. corymbiae* Crous and Summerell, *P. corymbiicola* Crous, *P. dipterocarpi* Suwannarach, and Lumyong, *P. dipterocarpicola* X. Tang, R.S. Jayawardena, *P. eucalypti*, *P. mangiferae* Dayarathne, Phookamsak, and K.D. Hyde, *P. myracrodruonis* A.P.S.L. Pádua, T.G.L. Oliveira, Souza-Motta, and J.D.P. Bezerra, *P. oldii* Cheew., M.J. Wingf. and Crous, and *P. variabile* Cheew., M.J. Wingf. and Crous in the Index Fungorum (accession date: 6 December 2022). The family was introduced in both asexual and sexual morphs. The sexual morph is characterized by immersed, beaked, ostiole ascomata, unitunicate asci, a non-amyloid subapical ring, hyaline ascospores that are 1-septate near the middle or aseptate, with terminal, elongated hyaline appendages. The asexual morph is characterized by superficial and immersed conidiomata with masses of apically proliferous conidiogenous cells and hyaline, ellipsoidal conidia, but with no conidiophores [1,2,3].

*Pseudoplagiostoma* species were mainly reported as endophytes, plant pathogens, or saprobes in various regions, viz. Asia, North America, Oceania, and South America [1,2,3,4,5,6,7,8,9]. As a type species, *P. eucalypti* was reported with more than 40 strains in the whole world (NCBI Nucleotide database, https://www.ncbi.nlm.nih.gov/nucleotide/, accessed on 6 December 2022). More than half of the *P. eucalypti* strains were distributed in Asia, including China, Malaysia, Thailand, and Viet Nam. *Pseudoplagiostoma eucalypti*, *P. oldii* and *P. variabile* possessed host preferences, and they almost occurred on *Eucalyptus* [1,10]. Recently, *Pseudoplagiostoma* as an endophyte from *Castanea mollissima* and *Dipterocarpus* sp. was introduced by Mu et al. [2] and Tang et al. [9]. This was the first time that *Pseudoplagiostoma* species had been found on the host of *Castanea* and *Dipterocarpus*.

The classifications were initially based on phenotype, and with the development of molecular technology, phylogenetic analysis of multi-gene provided reliable evidence for the classifications of phenotype [11,12,13]. However, this has led to significant changes in many lineages, and many unsuitable introductions of secondary ranking. Recently, Hyde et al. [13] used ‘temporal banding’ to revalued the position of higher taxa in the *Ascomycota* Caval.-Sm. They believed that the taxa of higher hierarchical levels should be older than lower levels. Thus, ‘temporal banding’ was regarded as a novel approach, using molecular clock analyses to standardize taxonomic ranking [11,13,14,15,16,17]. The concept of molecular clock studies is evaluating divergence times of lineages based on the assumption that mutations occur at balanced rate over time, and gradually become a reliable tool to calculate evolutionary events and explore new insights into genetic evolution [18,19,20]. Moreover, Hyde et al. [13] proposed a series of evolutionary periods including, families: 50–150 Mya, orders: 150–250 Mya, subclasses: 250–300 Mya, classes: 300–400 Mya, subphyla: 400–550 Mya, phyla > 550 Mya, and provided recommendations for ranking taxa with evidence for divergence times. The key to draw conclusions from divergence data was stabilize the phylogenetic trees.

In this article, three new species were described by combining phylogeny and morphology, viz. *Pseudoplagiostoma alsophilae* sp. nov., *P. bambusae* sp. nov., and *P. machili* sp. nov. At the same time, a hypothesis for specific divergence time and origin of *Pseudoplagiostomataceae* was proposed.

## 2. Materials and Methods

### 2.1. Isolation and Morphology

Diseased leaves of *Alsophila spinulosa* (Wall. ex Hook.) R. M. Tryon, Bambusoideae sp., *Machilus nanmu* (Oliver) Hemsley were collected from Fujian and Hainan Province during 2021 and 2022 in China. The cultures of *Pseudoplagiostomataceae* were isolated from diseased and non-diseased tissues of sample leaves using tissue isolation methods [21]. The diseased leaves with obvious disease spots were selected as experimental materials, and the surfaces of the materials were cleaned with sterile deionized water. The leaf samples with typical spot symptoms were first surface sterilized for 30 s in 75% ethanol, then rinsed in sterile deionized water for 45 s, in 2.5% sodium hypochlorite solution for 2 min, then rinsed four times in sterile deionized water for 45 s [22]. The pieces were blotted on sterile filter paper to dry, then transferred onto the PDA flats (PDA medium: potato 200 g, agar 15–20 g, dextrose 15–20 g, deionized water 1 L, pH ~7.0, available after sterilization), and incubated at 23 °C for 3–5 days. Hyphal tips were then removed to new PDA flats to gain pure cultures Simultaneously, inoculate on Petri dishes containing pine needle agar (PNA) [23], and incubated at 23 °C under continuous near ultraviolet light to promote sporulation.

After 10–14 days of incubation, morphological characters should be recorded, including graphs of the colonies were taken at the 10th and 14th day using a digital camera (Canon G7X), morphological characters of conidiomata using a stereomicroscope (Olympus SZX10), and micromorphological structures were observed using a microscope (Olympus BX53). All cultures were deposited in 10% sterilized glycerin and sterile water at 4 °C for future studies. Micromorphological structural measurements were taken using the Digimizer software (https://www.digimizer.com/, accessed on 6 December 2022), with 25 measurements taken for each structure [22]. Voucher specimens were deposited in the Herbarium Mycologicum Academiae Sinicae, Institute of Microbiology, Chinese Academy of Sciences, Beijing, China (HMAS), and Herbarium of the Department of Plant Pathology, Shandong Agricultural University, Taian, China (HSAUP). Ex-holotype living cultures were deposited in the Shandong Agricultural University Culture Collection (SAUCC). Taxonomic information of the new taxa was submitted to MycoBank (http://www.mycobank.org, accessed on 6 December 2022).

### 2.2. DNA Extraction and Amplification

Genomic DNA was extracted from fungal mycelia grown on PDA, using a kit (OGPLF-400, GeneOnBio Corporation, Changchun, China) according to the manufacturer’s protocol [24]. Gene sequences were obtained from five loci including the internal transcribed spacer regions with the intervening 5.8S nrRNA gene (ITS), the partial large subunit nrRNA gene (LSU), the partial DNA-directed RNA polymerase II subunit two gene (*rpb2*), the partial translation elongation factor 1-alpha gene (*tef1α*), and the partial beta-tubulin gene (*tub2*) were amplified by the primer pairs and polymerase chain reaction (PCR) programs listed in Table 1. Amplification reactions were performed in a 20 μL reaction volume, which contained 10 μL 2 × Hieff Canace^®^ Plus PCR Master Mix (With Dye) (Yeasen Biotechnology, Cat No. 10154ES03), 0.5 μL of each forward and reverse primer (10 μM) (TsingKe, Qingdao, China), and 1 μL template genomic DNA, adjusted with distilled deionized water to a total volume of 20 μL. PCR amplification products were visualized on 2% agarose electrophoresis gel. DNA Sequencing was performed using an Eppendorf Master Thermocycler (Hamburg, Germany) at the Tsingke Company Limited (Qingdao, China) bi-directionally. Consensus sequences were obtained using MEGA 7.0 [25]. All sequences generated in this study were deposited in GenBank (Table 2).

### 2.3. Phylogenetic Analyses

Novel sequences obtained in this study and related sets of sequences from Mu et al. [2] were aligned with MAFFT v. 7 and corrected manually using MEGA 7 [33]. Multi-locus phylogenetic analyses were based on the algorithms maximum likelihood (ML) and Bayesian inference (BI) methods. The ML was run on the CIPRES Science Gateway portal (https://www.phylo.org, accessed on 6 December 2022) [34] using RaxML–HPC2 on XSEDE v. 8.2.12 [35] and employed a GTRGAMMA substitution model with 1000 bootstrap replicates. Other parameters were default. For Bayesian inference analyses, the best model of evolution for each partition was determined using Modeltest v. 2.3 [36] and included the analyses. The BI was performed in MrBayes on XSEDE v. 3.2.7a [37,38,39], and two Markov chain Monte Carlo (MCMC) chains were run, starting from random trees, for 2,000,000 generations. Additionally, sampling frequency of 100th generation. The first 25% of trees were discarded as burn-in, and BI posterior probabilities (PP) were conducted from the remaining trees. The consensus trees were optimized using FigTree v. 1.4.4 (http://tree.bio.ed.ac.uk/software/figtree, accessed on 6 December 2022), and embellished with Adobe Illustrator CC 2019 (Figure 1).

### 2.4. Divergence Time Estimation

An ITS + LSU + *rpb2* + *tef1α* + *tub2* sequence dataset with 54 strains was used to infer the divergence times of species in the family *Pseudoplagiostomataceae* (Figure 2). An XML file was conduct with BEAUti v. 2 and run with BEAST v. 2.6.5. The rates of evolutionary changes at nuclear acids were estimated using MrModeltest v. 2.3 with the GTR substitution model [36,40]. Divergence time and corresponding CIs were taken with a Relaxed Clock Log Normal and the Yule speciation prior. Three fossil time points, i.e., *Protocolletotrichum deccanense* [41], *Spataporthe taylorii* [42], and *Paleopyrenomycites devonicus* [43,44], representing the divergence time at *Capnodiales*, *Diaporthales*, and *Pezizomycotina* were selected for calibration, respectively. The offset age with a gamma distributed prior (scale = 20 and shape = 1) was set as 65, 136, and 400 Mya for *Colletotrichum*, *Diaporthales*, and *Pezizomycotina*, respectively. After 100,000,000 generations, the first 20% were removed as burn in. Convergence of the log file was checked for with Tracer v. 1.7.2 (ESS > 200 was considered convergence). Afterwards, a maximum clade credibility (MCC) tree was integrated with TreeAnnotator v. 2.6.5, and annotating clades with posterior probability (PP) > 0.7.

### 2.5. Inferring Historical Biogeography

The Reconstruct Ancestral State in Phylogenies (RASP) v. 4.2 was used to reconstruct historical biogeography for the family *Pseudoplagiostomataceae* [45,46]. Maximum clade credibility (MCC) tree, consensus tree, and states were checked with RASP before analysis. Based on the results, we select the Statistical Dispersal–Extinction–Cladogenesis (S–DEC) model. The geographic distributions for *Pseudoplagiostomataceae* were identified in four areas: (A) Asia, (B) Oceania, (C) South America, and (D) North America.

## 3. Results

### 3.1. Phylogenetic Analyses

Alignment contained 25 strains representing *Pseudoplagiostomataceae* and *Apoharknessiaceae*, and the strain CBS 243.76 of *Nakataea oryzae* was used as outgroup. The dataset had an aligned length of 3343 characters including gaps were obtained, viz. LSU: 1–842, ITS: 843–1544, *rpb2*: 1545–2215, *tef1α*: 2216–2813, *tub2*: 2814–3343 (Appendix A). Of these, 2059 were constant, 303 were parsimony-uninformative, and 981 were parsimony-informative. The ModelTest suggested that the BI used the Dirichlet base frequencies, and the GTR + I + G evolutionary mode for LSU, ITS, and *tub2*, GTR + I for *rpb2*, and HKY + G for *tef1α*. The topology of the ML tree was consistent with that of the Bayesian tree, and, therefore, only shown the topology of the ML tree as a representative for recapitulating evolutionary relationship within the family *Pseudoplagiostomataceae*. The final ML optimization likelihood was −14,845.00184. The 25 strains were assigned to 18 species clades on the phylogram (Figure 1). Based on the phylogenetic resolution and morphological analyses, the present study introduced three novel species of the *Pseudoplagiostomataceae*, viz. *Pseudoplagiostoma alsophilae* sp. nov., *P. bambusae* sp. nov., and *P. machili* sp. nov.

### 3.2. Divergence Time Estimation for Pseudoplagiostomataceae

Divergence time estimation (Figure 2) showed that *Pseudoplagiostomataceae* occurred early with a mean stem age of 104.1 Mya [95% highest posterior density (HPD) of 86.0–129.0 Mya, 1.0 PP], and a mean crown age of 91.6 Mya (95% HPD of 73.4–117.6 Mya, 0.9 PP), which was consistent with a previous study [13]. The clade of *Pseudoplagiostoma eucalypti* and *P. oldii* with a mean stem age of 10.7 Mya (95% HPD of 4.9–20.9 Mya), and a mean crown age of 4.6 Mya (95% HPD of 1.5–9.7 Mya), which was consistent with previous studies [47]. While the clade of *Pseudoplagiostoma eucalypti* and *P. oldii* evolved most recently, the clade of *P. myracrodruonis* and *P. castaneae* diverged the earliest in the genus with a stem age of 68.1 Mya (95% HPD of 39.7–98.8 Mya). The stem/crown age of other species are shown in Table 3.

### 3.3. The Historical Biogeography of Pseudoplagiostomataceae

Historical biogeography scenarios of *Pseudoplagiostomataceae* were inferred by RASP (Figure 3). The RASP analysis indicated that Asia is the original center of *Pseudoplagiostomataceae*, and suggests that five dispersal events (one from Asia to Oceania, one from Oceania to Asia, two from Oceania to South America, and one from Oceania to North America) and four vicariance (*Pseudoplagiostoma eucalypti*, *P. oldii*, *P. variabile*, *P. dipterocarpi* and *P. castaneae*) events emerged during the distribution of this genus (Figure 3a). Meanwhile, eight species were found in Asia, three in Oceania, three in South America, and one in North America, indicating that Asia is still the center of *Pseudoplagiostomataceae* species. Afterwards, a total of 42 specimens of *P. eucalypti* (twenty-five in Asia, seven in Oceania, nine in North America and one in South America) have been collected, suggesting that Asia is the ancestral area (Figure 4). Meanwhile, possible concealed dispersal routes were indicated (Figure 3b): (1) Asia to Oceania, (2) Oceania to North America and (3) Oceania to South America.

### 3.4. Taxonomy

#### 3.4.1. *Pseudoplagiostoma alsophilae* Z.X. Zhang, Z. Meng and X.G. Zhang, sp. nov.

MycoBank—No: MB846483

Etymology—The epithet “*alsophilae*” pertains to the generic name of the host plant *Alsophila spinulosa*.

Type—China. Hainan Province, Wuzhishan National Nature Reserve, on diseased leaves of *Alsophila spinulosa*, 20 May 2021, Z.X. Zhang, holotype HMAS 352298, ex-holotype living culture SAUCC WZ0451.

Description—Leaf is endogenic and associated with leaf spots. Sexual morph (PDA): Ascomata 300–450 × 300–400 μm, buried or attached to the surface of mycelia, aggregative or solitary, globose to elliptical, brown to black, exuding hyaline asci. Asci 60–110 × 12–19 μm, unitunicate, 8-spored, subcylindrical to long obovoid, wedge-shaped. Ascospores 19–24 × 8–10.5 μm, overlapping uni- to bi-seriate, lageniform, sharpening to apex, hyaline, median 1- septate. Asexual morph (PNA): Conidiomata pycnidial, growing on the surface of pine needles, globose to subglobose, 150–250 × 200–300 μm, solitary, black, exuding creamy yellow conidia. Conidiophores indistinct, often reduced to conidiogenous cells. Conidiogenous cells hyaline, smooth, multi-guttulate, cylindrical to ampulliform, attenuate towards apex, phialidic, 8–13 × 1.5–3 μm. Conidia aseptate, globose to irregular globose, broad ellipsoid, apex obtuse, base tapering, hyaline, smooth, guttulate, 17–21 × 13–15 μm (mean = 19.3 ± 1.2 × 14.2 ± 0.6 μm, n = 30), base with peg-like hila, 1.0–1.5 μm diam, see Figure 5.

Culture characteristics—Colonies on PDA flat at 23 °C for 14 days in dark reach 77–83 mm in diameter, grey-white to creamy white with irregular margin, spread like petals from the inside and outside, reverse is similar. Colonies on PNA flat at 23 °C for 14 days in dark reach 33–36 mm in diameter, white with regular margin, with slight aerial mycelia, reverse is similar.

Additional specimen examined—China. Hainan Province, Wuzhishan National Nature Reserve, on dead leaves of a broadleaf tree, 20 May 2021, Z.X. Zhang, HSAUP WZ0152, living culture SAUCC WZ0152.

Notes—Phylogenetic analyses of five combined genes (LSU, ITS, *rpb2*, *tef1α* and *tub2*) showed *Pseudoplagiostoma alsophilae* sp. Nov. formed an independent clade and was closely related to *P. dipterocarpi*, *P. dipterocarpicola,* and *P. mangiferae* (Figure 1). In detail, *P. alsophilae* is distinguished from *P. dipterocarpi* by 50/507 bp in ITS and 21/838 in LSU, from *P. dipterocarpicola* by 57/600 in ITS, 8/820 in LSU, 67/211 in *tef1α* and 96/481 in *tub2*, and from *P. mangiferae* by 64/573 in ITS and 10/778 in LSU. The morphological characteristics of *P. alsophilae* differing from *P. dipterocarpi*, *P. dipterocarpicola,* and *P. mangiferae* are listed in Table 4 [3,8,9].

#### 3.4.2. *Pseudoplagiostoma bambusae* Z.X. Zhang, Z. Meng, and X.G. Zhang, sp. nov.

MycoBank—No: MB846484

Etymology—The epithet “*bambusae*” pertains to the host plant Bambusoideae.

Type—China. Fujian Province, Fujian Wuyi Mountain National Nature Reserve, on diseased leaves of Bambusoideae sp., 15 October 2022, Z.X. Zhang, holotype HMAS 352300, ex-holotype living culture SAUCC 1206-4.

Description—Leaf is endogenic and associated with leaf spots. Conidiomata pycnidial, aggregated or solitary, globose to irregular, 200–250 × 150–250 μm, formed on agar surface, slimy, black, semi-submerged, exuding hyaline conidia. Conidiophores indistinct, often reduced to conidiogenous cells. Conidiogenous cells hyaline, smooth, cylindrical to ampulliform, attenuate towards apex, phialidic, 5–13 × 1.5–2.5 μm. Conidia aseptate, oblong to broad ellipsoid, base tapering, hyaline, smooth, guttulate, slightly depressed in the middle, 13–20 × 5.7–7.6 μm (mean = 15.2 ± 1.6 × 6.7 ± 0.5 μm, n = 30), base with inconspicuous to conspicuous hilum, 1.0–1.3 μm diam, see Figure 6. Sexual morph: unknown.

Culture characteristics—Colonies on PDA flat at 23 °C for 14 days in dark reach 43–48 mm in diameter, bluish-green to grey-white, with moderate aerial mycelia and undulate margin, reverse is similar.

Additional specimen examined—China. Fujian Province, Fujian Wuyi Mountain National Nature Reserve, on diseased leaves of Bambusoideae sp., 15 October 2022, Z.X. Zhang, HSAUP 1206-6, living culture SAUCC 1206-6.

Notes—Phylogenetic analyses of five combined genes showed *Pseudoplagiostoma bambusae* sp. nov. formed an independent clade and was closely related to *P. alsophilae* and *P. machili* (Figure 1). In detail, *P. bambusae* is distinguished from *P. alsophilae* by 48/613 bp in ITS, 12/828 in LSU, 144/535 in *tef1α* and 53/477 in *tub2*, and from *P. machili* by 67/615 in ITS, 9/828 in LSU, 156/536 in *tef1α* and 71/485 in *tub2*. Morphologically, *P. bambusae* differs from *P. alsophilae* and *P. machili* in several characteristics, as shown in Table 4.

#### 3.4.3. *Pseudoplagiostoma machili* Z.X. Zhang, Z. Meng, and X.G. Zhang, sp. nov.

MycoBank No: MB846485

Etymology—The epithet “*machili*” pertains to the generic name of the host plant *Machilus nanmu*.

Type—China. Hainan Province, Bawangling National Forest Park, on diseased leaves of *Machilus nanmu*, 19 May 2021, Z.X. Zhang, holotype HMAS 352299, ex-holotype living culture SAUCC BW0233.

Description—Leaf is endogenic and associated with leaf spots. Conidiomata pycnidial, aggregated or solitary, globose to irregular, 150–200 × 100–250 μm, black, exuding yellow conidia. Conidiophores indistinct, often reduced to conidiogenous cells. Conidiogenous cells hyaline, smooth, cylindrical to ampulliform, attenuate towards apex, phialidic, 7–16 × 2–3.5 μm. Conidia aseptate, ellipsoid to broad ellipsoid, apex obtuse, base tapering, hyaline, smooth, guttulate, 17.5–23 × 10.5–13.5 μm (mean = 20.7 ± 1.6 × 12.4 ± 0.7 μm, n = 30), base with inconspicuous to conspicuous hilum, 1.3–1.5 μm diam, see Figure 7. Sexual morph: unknown.

Culture characteristics—Colonies on PDA flat at 23 °C for 14 days in dark reach 58–62 mm in diameter, grey-white to creamy white, with moderate aerial mycelia and undulate margin, reverse is similar.

Additional specimen examined—China. Hainan Province, Bawangling National Forest Park, on diseased leaves of *Machilus nanmu*, 19 May 2021, Z.X. Zhang, HSAUP BW0221, living culture SAUCC BW0221.

Notes—Based on phylogeny and morphology, strains SAUCC BW0233 and SAUCC BW0221 were identified to the same species *Pseudoplagiostoma machili* sp. nov. For details, please refer to the notes for *Pseudoplagiostoma bambusae*.

## 4. Discussion

In the present study, three new species (*Pseudoplagiostoma alsophilae*, *P. bambusae*, and *P. machili*) from three hosts (*Alsophila spinulosa*, Bambusoideae sp., *Machilus nanmu*) in two provinces of China were illustrated and described (Figure 5, Figure 6 and Figure 7). *P. alsophilae* reproduced both asexually and sexually, while *P. bambusae* and *P. machili* only reproduced asexually. Most species of *Pseudoplagiostomataceae* were isolated from *Eucalyptus* (*Myrtaceae*) (*Pseudoplagiostoma corymbiae*, *P. corymbiicola*, *P. eucalypti*, *P. oldii*, and *P. variabile*), especially *P. eucalypti* with more than 40 strains [1,4,5,10]. Recently, other hosts were reported, including *Anacardiaceae* (*P. mangiferae* and *P. myracrodruonis*), *Dipterocarpaceae* (*P. dipterocarpi* and *P. dipterocarpicola*), *Fagaceae* (*P. castaneae*) [2,3,7,8,9]. This study puts more families in the host list, and they are *Cyatheaceae* (*P. alsophilae*), *Gramineae* (*P. bambusae*), and *Lauraceae* (*P. machili*). It has significant research value in regional species diversity and ecological diversity.

Currently, the divergence and ranking of taxa across the kingdom Fungi, especially the phylum *Ascomycota*, have significant theoretical and practical significance, and gradually become a reliable and referential evidence before introducing new higher taxa [11,13,14,15,16,17]. Our analysis of molecular clock indicates that *Pseudoplagiostomataceae* was closely related to *Apoharknessiaceae*, which was most deeply diverged during the Paleogene, with a mean stem age of 104.1 Mya (95% HPD of 86.0–129.0 Mya), and full supports (1.0 PP, Figure 2 and Table 3). Even though Hyde et al. [13] only included two species of the *Pseudoplagiostomataceae*, its divergence time was coincided with this study. In the present study, a mean stem age of *Diaporthales* reached 188.2 Mya and was fully supported earlier than in the previous study [13,47]. Therefore, both new fossil findings and new species findings have an impact on the divergence time of the orders. Of course, the impact was controllable, and it must be in certain evolutionary periods.

Macrofungi have been widely applied for biogeographical analyses [24,48,49,50,51]. Our study suggested that the species distribution and speciation of *Pseudoplagiostomataceae* had a particular biogeographical pattern, and these species appeared to originate in Asia, particularly in Southeast Asia. Previous studies suggested that the Indian continent collided with the Eurasian continent at ~60 Mya, which was consistent with some speciation of the *Pseudoplagiostomataceae*, and formed the Hengduan–Himalayan area which was a global biodiversity hotspot [52,53,54,55,56,57]. Based on the discovered specimens and biogeographical information, this study is more inclined to explain that *Pseudoplagiostomataceae* species originated in Asia and spread to Hawaii and South America through Malaysia, Australia, New Zealand, and more than 20,000 independent islands in the South Pacific, and frequent hurricanes and circulating ocean currents in the South Pacific are the best spore carriers. The humid climate in the southern hemisphere and the rich tropical host plants, such as *Quercus* sp. and *Eucalyptus* sp., are also suitable for the reproduction and evolution of *Pseudoplagiostomataceae* species [58,59]. Dispersal, vicariance, and extinction of species may be related to the Indian continent collided with the Eurasian; however, this claim needs more species and fossil evidence to support it.

## Figures and Tables

**Figure 1 jof-09-00082-f001:**
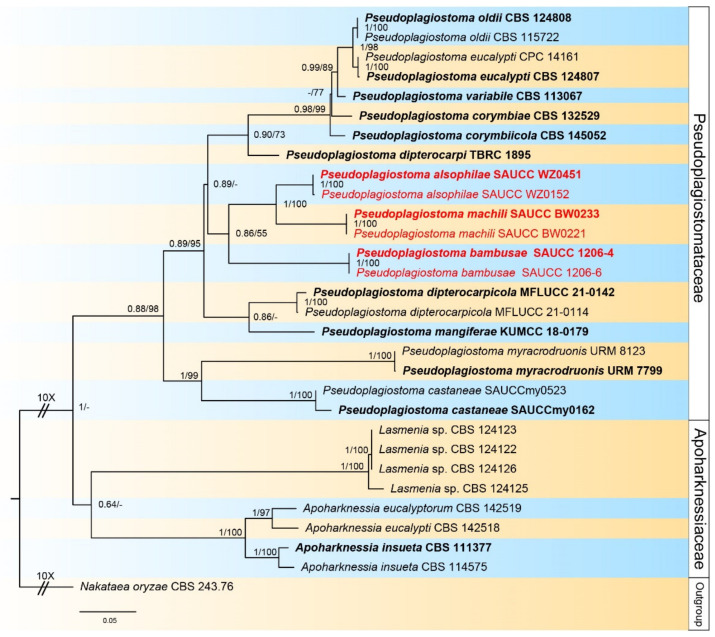
A phylogram of the *Pseudoplagiostomataceae* and *Apoharknessiaceae*, based on a concatenated ITS, LSU, *rpb2*, *tef1α*, and *tub2* sequence alignment, with *Nakataea oryzae* (CBS 243.76) as outgroup. BI posterior probabilities and maximum likelihood bootstrap support values above 0.60 and 50% are shown at the first and second position, respectively. Ex-type cultures are marked in bold face. Strains obtained in the present study are in red. Some branches are shortened for layout purposes—these are indicated by two diagonal lines with the number of times. The scale bar at the left–bottom represents 0.05 substitutions per site.

**Figure 2 jof-09-00082-f002:**
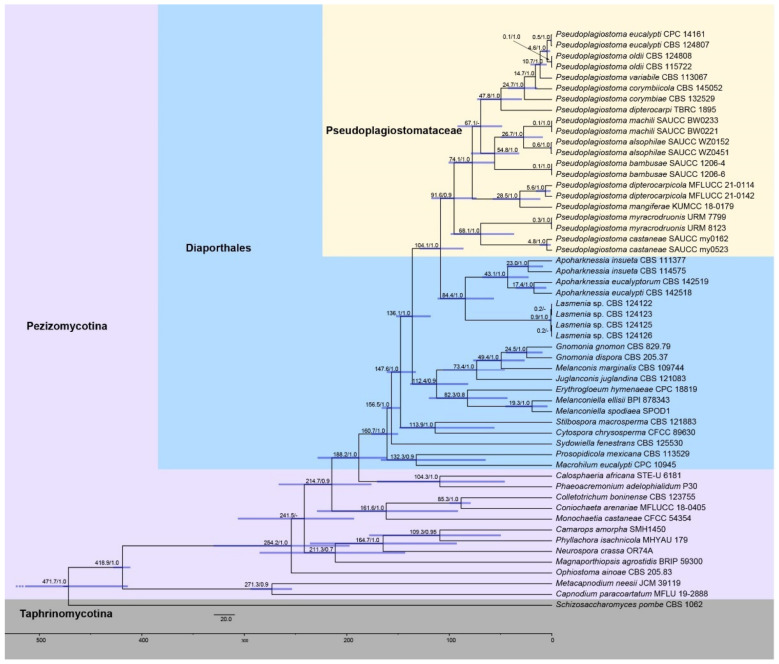
An estimated divergence of *Pseudoplagiostomataceae* generated from molecular clock analyses using a combined dataset of ITS, LSU, *rpb2*, *tef1α*, and *tub2* sequences. Estimated mean divergence time (Mya) and posterior probabilities (PP) > 0.7 are annotated at the internodes. The 95% highest posterior density (HPD) interval of divergence time estimates is marked by horizontal blue bars.

**Figure 3 jof-09-00082-f003:**
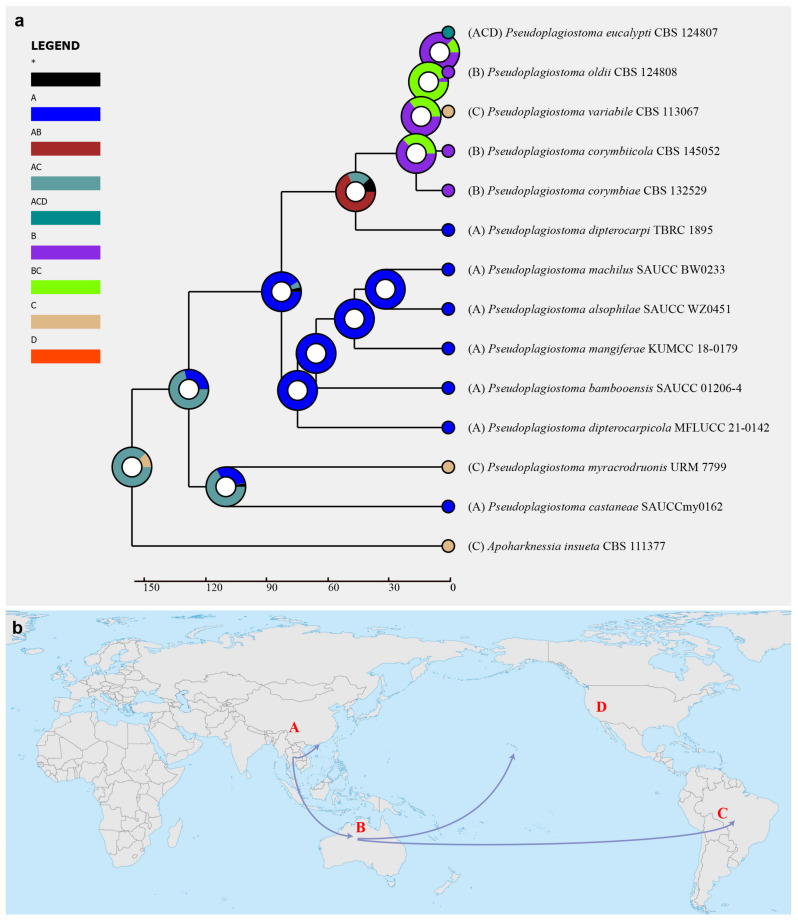
(**a**) Ancestral state reconstruction and divergence time estimation of *Pseudoplagiostomataceae* using a dataset containing ITS, LSU, *rpb2*, *tef1α*, and *tub2* sequences. A pie chart at each node suggested the possible ancestral distributions deduced from Statistical Dispersal–Extinction–Cladogenesis (S–DEC) analysis completed in RASP. A black asterisk stands for other ancestral ranges. (**b**) Possible dispersal routes of *Pseudoplagiostomataceae*. Areas were marked as follows: (A) Asia, (B) Oceania, (C) South America, (D) North America, (A,B) Asia and Europe, (A,C) Asia and South America, (B,C) Oceania and South America, and (A,C,D) Asia, South America, and North America.

**Figure 4 jof-09-00082-f004:**
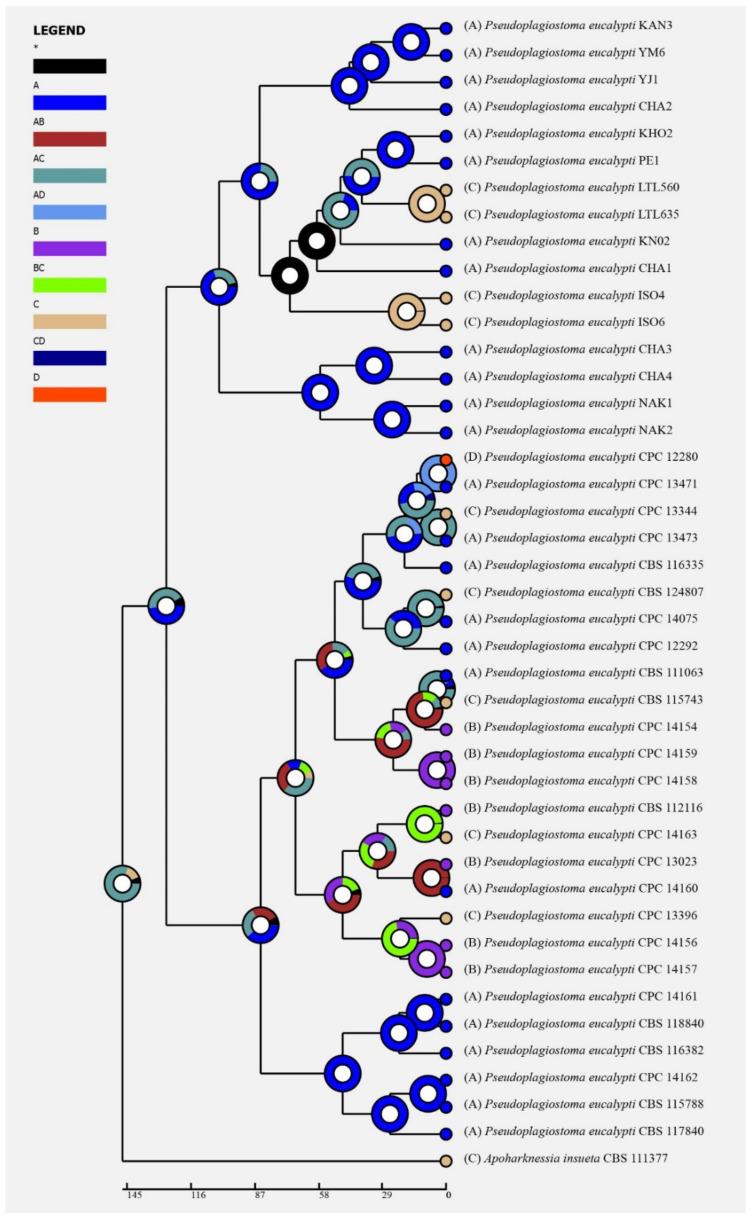
Ancestral state reconstruction and divergence time estimation of *Pseudoplagiostoma eucalypti* using a dataset containing ITS, LSU, *rpb2*, *tef1α*, and *tub2* sequences. A pie chart at each node indicates the possible ancestral distributions deduced from Statistical Dispersal–Extinction–Cladogenesis (S–DEC) analysis completed in RASP. A black asterisk stands for other ancestral ranges. Areas were marked as follows: (A) Asia, (B) Oceania, (C) South America, (D) North America, (A,B) Asia and Europe, (A,C) Asia and South America, (B,C) Oceania and South America, and (A,C,D) Asia, South America, and North America.

**Figure 5 jof-09-00082-f005:**
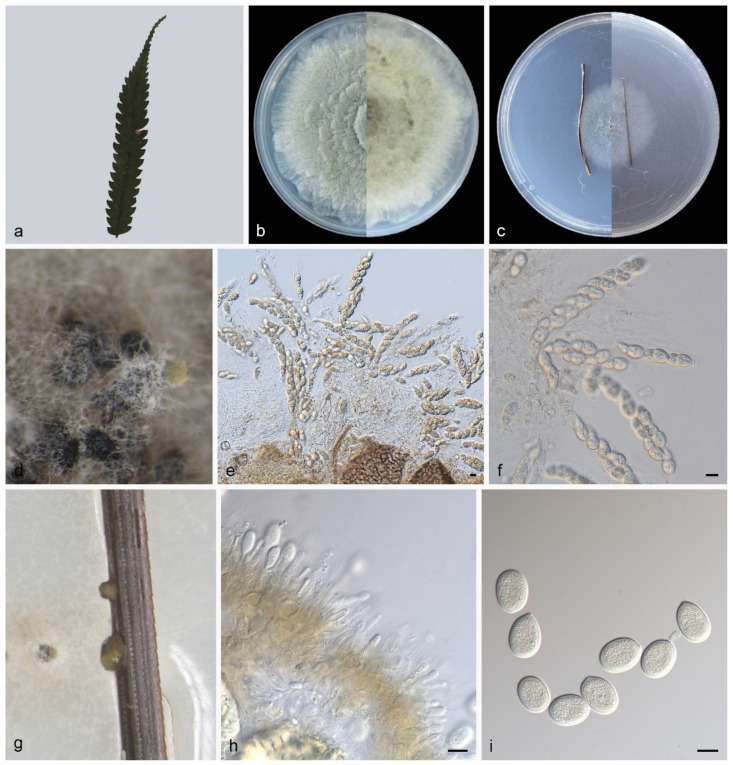
*Pseudoplagiostoma alsophilae* (holotype HMAS 352298). (**a**), leaves of host plant; (**b**,**c**), (left-above, right-reverse) after 15 days on PDA (**b**) and PNA (**c**); (**d**,**g**), colony overview; (**e**,**f**), asci and ascospores; (**h**,**i**), conidiogenous cells with conidia. Scale bars: (**e**,**f**,**h**,**i**), 10 μm.

**Figure 6 jof-09-00082-f006:**
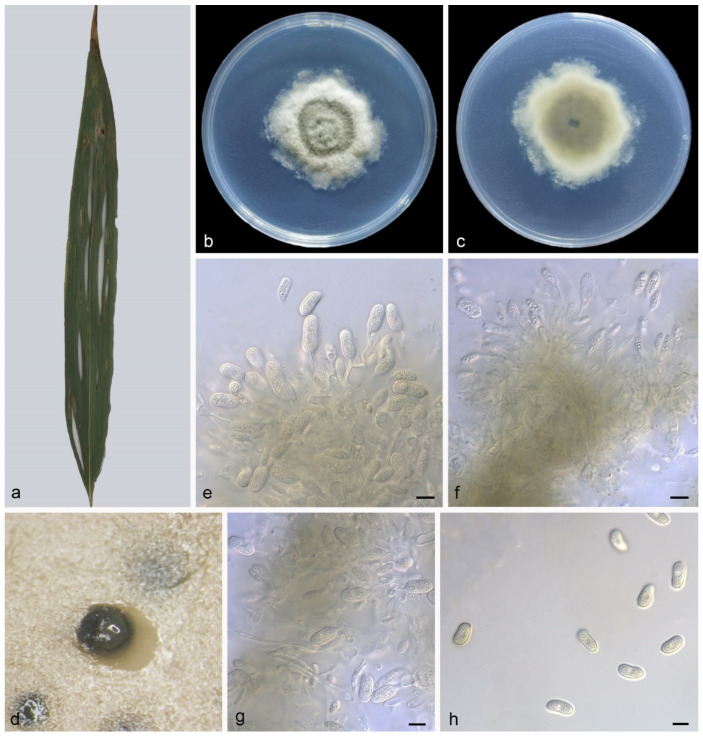
*Pseudoplagiostoma bambusae* (holotype HMAS 352300). (**a**) leaves of host plant; (**b**,**c**) inverse and reverse sides of colony after 15 days on PDA; (**d**) colony overview; (**e**–**g**) conidiogenous cells with conidia; (**h**) conidia. Scale bars: (**e**–**h**), 10 μm.

**Figure 7 jof-09-00082-f007:**
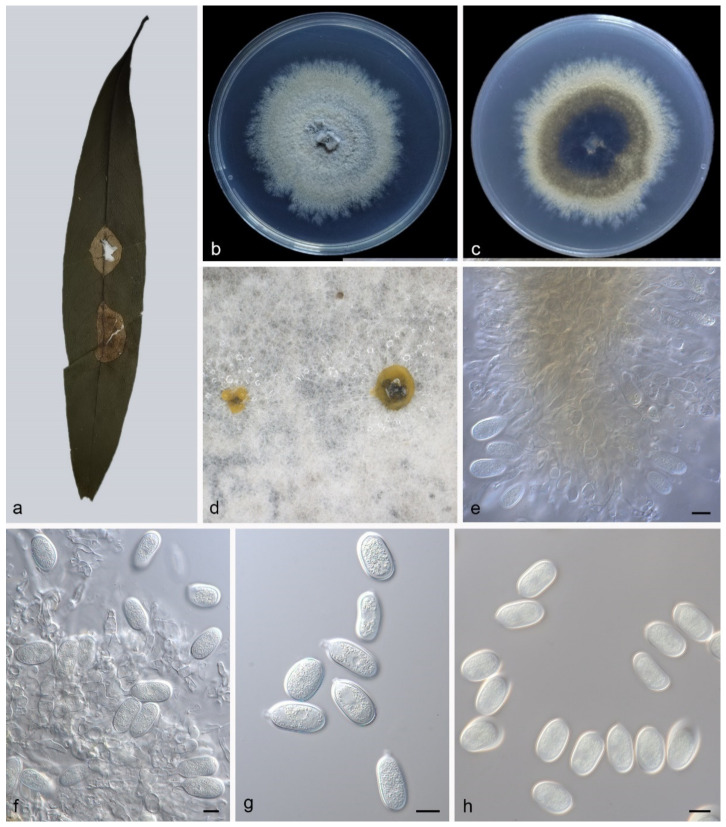
*Pseudoplagiostoma machili* (holotype HMAS 352299). (**a**) leaves of host plant; (**b**,**c**) inverse and reverse sides of colony after 15 days on PDA; (**d**) colony overview; (**e**,**f**), Conidiogenous cells with conidia; (**g**,**h**) conidia. Scale bars: (**e**–**h**) 10 μm.

**Table 1 jof-09-00082-t001:** Molecular markers and their PCR primers and programs used in this study.

Loci	PCR Primers	Sequence (5′—3′)	PCR Cycles	References
ITS	ITS5 ITS4	GGA AGT AAA AGT CGT AAC AAG G TCC TCC GCT TAT TGA TAT GC	(95 °C: 30 s, 55 °C: 30 s, 72 °C: 1 min) × 35 cycles	[26]
LSU	LR0R LR5	GTA CCC GCT GAA CTT AAG C TCC TGA GGG AAA CTT CG	(95 °C: 30 s, 52 °C: 30 s, 72 °C: 1 min) × 35 cycles	[27,28]
*rpb2*	fRPB2-5F fRPB2-7R	GAY GAY MGW GAT CAY TTY GG CCC ATW GCY TGC TTM CCC AT	(95 °C: 30 s, 56 °C: 30 s, 72 °C: 1 min) × 35 cycles	[29]
*tef1α*	EF1-728F EF-2	CAT CGA GAA GTT CGA GAA GG GGA RGT ACC AGT SAT CAT GTT	(95 °C: 30 s, 48 °C: 30 s, 72 °C: 1 min) × 35 cycles	[30,31]
*tub2*	Bt-2a Bt-2b	GGT AAC CAA ATC GGT GCT GCT TTC ACC CTC AGT GTA GTG ACC CTT GGC	(95 °C: 30 s, 53 °C: 30 s, 72 °C: 1 min) × 35 cycles	[32]

**Table 2 jof-09-00082-t002:** Information of specimens used in this study.

Fungal Species	Voucher	Substrate	Country	GenBank Accession				
				ITS	LSU	*tef1α*	*tub2*	*rpb2*
*Apoharknessia eucalypti*	CBS 142518	*Eucalyptus pellita*	Malaysia	MG934432	MN162172	–	MG934505	–
*A. eucalyptorum*	CBS 142519	*Eucalyptus pellita*	Malaysia	KY979752	KY979807	–	KY979919	–
*A. insueta*	CBS 111377 *	*Eucalyptus pellita*	Brazil	JQ706083	AY720814	MN271820	–	–
	CBS 114575	*Eucalyptus pellita*	Brazil	MN172402	MN172370	MN271821	–	–
*Calosphaeria africana*	STE-U 6181	*Prunus armeniaca*	South Africa	EU367445	EU367455	–	EU367465	–
*Camarops amorpha*	SMH1450	–	Puerto Rico		AY780054	–	AY780093	AY780156
*Capnodium paracoartatum*	MFLU 19-2888	*Ficus* sp.	Thailand	MT177926	MT177953	–	–	–
*Colletotrichum boninense*	CBS 123755	*Crinum asiaticum*	Japan	MH863323	MH874855	–	JQ005588	–
*Coniochaeta arenariae*	MFLUCC 18-0405 *	*Ammophila arenaria*	UK	MN047126	MN017896	–	–	–
*Cytospora chrysosperma*	CFCC 89630	*Salix psammophila*	China	KF765674	KF765690	–	–	KF765706
*Erythrogloeum hymenaeae*	CPC 18819	*Hymenaea courbaril*	Brazil	JQ685519	JQ685525	–	–	–
*Gnomonia dispora*	CBS 205.37	–	Netherlands	MH855886	MH867397	–	–	–
*G. gnomon*	CBS 829.79	*Populus* sp.	Switzerland	AY818957	AY818964	EU221905	EU219172	–
*Juglanconis juglandina*	CBS 121083	*Juglans regia*	Austria	KY427148	KY427148	KY427217	–	KY427198
*Lasmenia* sp.	CBS 124122	*Nephelium lappaceum*	Puerto Rico	GU797405	JF838337	–	–	–
	CBS 124123	*Nephelium lappaceum*	Puerto Rico	GU797406	JF838338	–	–	–
	CBS 124124	*Nephelium lappaceum*	Puerto Rico	JF838336	JF838341	–	–	–
	CBS 124125	*Nephelium lappaceum*	Puerto Rico	GU797407	JF838340	–	–	–
*Macrohilum eucalypti*	CPC 10945	*Eucalyptus* sp.	New Zealand	DQ195781	DQ195793	–	–	–
*Magnaporthiopsis agrostidis*	BRIP 59300	*Agrostis stolonifera*	Australia	KT364753	KT364754	KT364756	–	–
*Melanconiella ellisii*	BPI 878343	*Carpinus caroliniana*	USA	JQ926271	JQ926271	JQ926406	–	JQ926339
*M. spodiaea*	SPOD1	*Carpinus betulus*	Austria	JQ926301	JQ926301	JQ926434	–	JQ926367
*Melanconis marginalis*	AR 3442	*Alnus rubra*	Canada	EU199197	AF408373	EU221991	EU219103	EU219301
*Metacapnodium neesii*	JCM 39119	–	Japan	LC576698	LC576694	LC576697	–	LC576696
*Monochaetia castaneae*	CFCC 54354 *	*Castanea mollissima*	China	MW166222	MW166263	MW199741	MW218515	MW199737
*Nakataea oryzae*	CBS 243.76	*Oryza sativa*	Italy	MH860975	DQ341498	–	–	KM485077
*Neurospora crassa*	OR74A	–	India	HQ271348	AF286411	XM959775	–	AF107789
*Ophiostoma ainoae*	CBS 205.83	–	Norway	MH861571	MH873301	–	–	–
*Phaeoacremonium adelophialidum*	P30	*Vitis vinifera*	Algeria	MW689543	MW689544	–	–	–
*Phyllachora isachnicola*	MHYAU 179	*Isachne albens*	China	MH018561	MH018563	–	–	–
*Prosopidicola mexicana*	CBS 113529	*Prosopis glandulosa*	Netherlands	AY720709	–	–	–	–
*Pseudoplagiostoma alsophilae*	SAUCC WZ0451 *	*Alsophila spinulosa*	China	OP810625	OP810631	OP828580	OP828586	OP828578
	SAUCC WZ0152	*Alsophila spinulosa*	China	OP810626	OP810632	OP828581	OP828587	OP828579
*P. bambusae*	SAUCC 1206-4 *	Bambusoideae sp.	China	OP810629	OP810635	OP828584	OP828590	–
	SAUCC 1206-6	Bambusoideae sp.	China	OP810630	OP810636	OP828585	OP828591	–
*P. castaneae*	SAUCCmy0162 *	*Castanea mollissima*	China	MZ156982	MZ156985	MZ220321	MZ220325	MZ220323
	SAUCCmy0523	*Castanea mollissima*	China	MZ156983	MZ156986	MZ220322	MZ220326	MZ220324
*P. corymbiae*	CBS 132529 *	*Corymbia* sp.	Australia	JX069861	JX069845	–	–	–
*P. corymbiicola*	CBS 145052 *	*Corymbia citriodora*	Australia	MK047425	MK047476	MK047558	MK047577	–
*P. dipterocarpi*	TBRC 1895 *	*Dipterocarpus tuberculatus*	Thailand	KR994682	KR994683	–	–	–
*P. dipterocarpicola*	MFLUCC 21-0142 *	*Dipterocarpus* sp.	Thailand	OM228844	OM228842	OM219629	OM219638	–
	MFLUCC 21-0114	*Dipterocarpus* sp.	Thailand	OM228843	OM228841	OM219628	OM219637	–
*P. eucalypti*	CBS 124807 *	*Eucalyptus urophylla*	Venezuela	GU973512	GU973606	GU973542	GU973575	–
	CPC 14161	*Eucalyptus camaldulensis*	Viet Nam	GU973510	GU973604	GU973540	GU973573	–
	KAN3	*Eucalyptus* sp.	Thailand	AB627948	–	–	–	–
	KHO2	*Eucalyptus* sp.	Thailand	AB630954	–	–	–	–
	CHA1	*Eucalyptus* sp.	Thailand	AB630955	–	–	–	–
	CHA2	*Eucalyptus* sp.	Thailand	AB630956	–	–	–	–
	CHA3	*Eucalyptus* sp.	Thailand	AB630957	–	–	–	–
	CHA4	*Eucalyptus* sp.	Thailand	AB630958	–	–	–	–
	NAK1	*Eucalyptus* sp.	Thailand	AB630959	–	–	–	–
	NAK2	*Eucalyptus* sp.	Thailand	AB630960	–	–	–	–
	KN02	*Eucalyptus pulverulenta*	Japan	AB978371	–	–	AB978372	–
	CPC 12280	*Eucalyptus* sp.	USA	GU973507	GU973601	GU973537	GU973570	–
	CBS 111063	–	Malaysia	GU973508	GU973602	GU973538	GU973571	–
	CPC 115743	*Eucalyptus globulus*	Uruguay	GU973509	GU973603	GU973539	GU973572	–
	CPC 13344	*Eucalyptus urophylla*	Venezuela	GU973511	GU973605	GU973541	GU973574	–
	CBS 112116	*Angophora* sp.	Australia	GU973513	GU973607	GU973543	GU973576	–
	CBS 116382	*Eucalyptus camaldulensis*	Thailand	GU973514	GU973608	GU973544	GU973577	–
	CPC 12292	*Eucalyptus camaldulensis*	Bhutan	GU973515	–	GU973545	GU973578	–
	CBS 118840	*Eucalyptus camaldulensis*	Thailand	GU973517	–	GU973547	GU973580	–
	CPC 14163	*Eucalyptus globulus*	Uruguay	GU973518	–	GU973548	GU973581	–
	CPC 14075	*Eucalyptus urophylla*	China	GU973519	–	GU973549	GU973582	–
	CBS 116335	*Eucalyptus camaldulensis*	Viet Nam	GU973520	–	GU973550	GU973583	–
	CPC 13023	*Eucalyptus longifolia*	Australia	GU973521	–	GU973551	GU973584	–
	CPC 14160	*Eucalyptus camaldulensis*	Viet Nam	GU973522	–	GU973552	GU973585	–
	CPC 13396	*Eucalyptus* sp.	Venezuela	GU973523	–	GU973553	GU973586	–
	CPC 14156	*Eucalyptus saligna*	Australia	GU973524	–	GU973554	GU973587	–
	CPC 14157	*Eucalyptus saligna*	Australia	GU973525	–	GU973555	GU973588	–
	CPC 14159	*Eucalyptus pellita*	Australia	GU973526	–	GU973556	GU973589	–
	CPC 14154	*Eucalyptus urophylla*	Australia	GU973527	–	GU973557	GU973590	–
	CPC 14158	*Eucalyptus pellita*	Australia	GU973528	–	GU973558	GU973591	–
	CPC 13471	*Eucalyptus camaldulensis*	Thailand	GU973529	–	GU973559	GU973592	–
	CPC 13473	*Eucalyptus camaldulensis*	Thailand	GU973530	–	GU973560	GU973593	–
	CPC 14162	*Eucalyptus camaldulensis*	Viet Nam	GU973531	–	GU973561	GU973594	–
	CBS 115788	*Eucalyptus camaldulensis*	Thailand	GU973532	–	GU973562	GU973595	–
	CBS 117840	*Eucalyptus camaldulensis*	Viet Nam	GU973533	–	GU973563	GU973596	–
	PE1	*Eucalyptus robusta*	China	KT831771	–		KT831772	–
	LTL560	*Eucalyptus microcorys*	Brazil	MF663591	–		–	–
	LTL635	*Eucalyptus microcorys*	Brazil	MF663594	–		–	–
	ISO4	*Eucalyptus grandis* x *Eucalyptus urophylla*	Brazil	MG832418	–	MG832416	–	–
	ISO6	*Eucalyptus grandis* x *Eucalyptus urophylla*	Brazil	MG832419	–	MG832417	–	–
	YJ1	–	China	MT801070	–	–	MT829072	–
	YM6	–	China	MT801071	–	–	MT829073	–
*P. mangiferae*	KUMCC 18-0179 *	*Mangifera* sp.	China	MK084824	MK084825	–	–	–
*P. myracrodruonis*	URM 7799 *	*Astronium urundeuva*	Brazil	MG870421	MK982151	MK982557	MN019566	MK977723
	URM 8123	*Astronium urundeuva*	Brazil	MK982150	MK982152	MK982558	MN019567	MK977724
*P. machili*	SAUCC BW0233 *	*Machilus nanmu*	China	OP810627	OP810633	OP828582	OP828588	–
	SAUCC BW0221	*Machilus nanmu*	China	OP810628	OP810634	OP828583	OP828589	–
*P. oldii*	CBS 115722	*Eucalyptus camaldulensis*	Australia	GU973535	GU973610	GU973565	GU993864	–
	CBS 124808 *	*Eucalyptus camaldulensis*	Australia	GU973534	GU973609	GU973564	GU993862	–
*P. variabile*	CBS 113067 *	*Eucalyptus globulus*	Uruguay	GU973536	GU973611	GU973566	GU993863	–
*Schizosaccharomyces pombe*	CBS 1062	–	Netherlands	KY105378	KY109602	–	–	–
*Stilbospora macrosperma*	CBS 121883	*Carpinus betulus*	Austria	JX517290	JX517299	–	–	KF570196
*Sydowiella fenestrans*	CBS 125530	*Chamerion angustifolium*	USA	JF681956	EU683078	–	–	–

Notes: Ex-type strains are marked with “*”. Novel species introduced are in bold in this study.

**Table 3 jof-09-00082-t003:** Inferred divergence time of species in the genus *Pseudoplagiostoma*.

Genus/Species	Means of Stem Age (Mya)/95% HPD (Mya)/Posterior Probabilities	Means of Crown Age (Mya)/95% HPD (Mya)/Posterior Probabilities
*Pseudoplagiostoma*	104.1/86.0–129.0/1.0	91.6/73.4–117.6/0.9
*P. alsophilae*	26.7/8.7–49.9/1.0	0.6/0.1–1.8/1.0
*P. bambusae*	55.8/31.8–79.0/1.0	0.1/0.1–0.8/1.0
*P. castaneae*	68.1/36.7–98.8/1.0	4.8/0.9–12.0/1.0
*P. corymbiae*	24.7/14.1–42.9/1.0	24.7/14.1–42.9/1.0
*P. corymbiicola*	6.0/7.6–27.4/1.0	6.0/7.6–27.4/1.0
*P. dipterocarpi*	47.8/29.1–73.0/1.0	47.8/29.1–73.0/1.0
*P. dipterocarpicola*	28.5/11.2–57.9/1.0	5.6/1.1–15.4/1.0
*P. eucalypti*	4.6/1.5–9.7/1.0	0.5/0.1–1.9/1.0
*P. machili*	26.7/8.7–49.9/1.0	0.1/0.1–0.7/1.0
*P. mangiferae*	28.5/11.2–57.9/1.0	28.5/11.2–57.9/1.0
*P. myracrodruonis*	68.1/36.7–98.8/1.0	0.3/0.1–1.3/1.0
*P. oldii*	4.6/1.5–9.7/1.0	0.1/0.1–0.8/1.0
*P. variabile*	10.7/5.0–21.0/1.0	10.7/5.0–21.0/1.0

**Table 4 jof-09-00082-t004:** Asexual morphological features of *Pseudoplagiostoma* species.

Species	Conidiogenous Cells	Size of Conidiogen-Ous Cells (μm)	Conidia	Size of Conidia (μm)	References
*P. alsophilae*	Cylindrical to ampulliform	8–13 × 1.5–3	Globose to irregular globose, broad ellipsoid	17–21 × 13–15	This study
*P. bambusae*	Cylindrical to ampulliform	5–13 × 1.5–2.5	Oblong to broad ellipsoid	13–20 × 5.7–7.6	This study
*P. castaneae*	Cylindrical to ampulliform	8–35 × 1–2	Ellipsoid, slightly curved	9–13.5 × 2–4.5	[2]
*P. corymbiae*	Cylindrical to ampulliform with long cylindrical neck	10–20 × 4–7	Elongate ellipsoidal	14–19 × 7–10	[4]
*P. corymbiicola*	Cylindrical to ampulliform with long cylindrical neck	15–30 × 3–5	Elongate ellipsoidal	15–20 × 6–8	[5]
*P. dipterocarpi*	Cylindrical to ampulliform	18–25 × 2.5–4.5	Elongate ellipsoidal	14–36 × 7–11	[3]
*P. dipterocarpicola*	Cylindrical to ampulliform	5–11 × 1–2.5	Ellipsoidal to elongated ellipsoidal	9–22 × 4–7.5	[9]
*P. eucalypti*	Cylindrical to ampulliform	6–15 × 2–6	Ellipsoidal	15–23 × 6.5–8.5	[1]
*P. machili*	Cylindrical to ampulliform	7–16 × 2–3.5	Ellipsoid to broad ellipsoid	17.5–23 × 10.5–13.5	This study
*P. mangiferae*	Cylindrical to ampulliform	5–11 × 3.2–12.6	Ellipsoidal	18–24 × 11–14	[8]
*P. myracrodruonis*	Lageniform to ampulliform	7–7.5 × 2–3.5	Ellipsoid, oblong-cylindrical	10–19 × 4–7.5	[7]
*P. oldii*	Cylindrical to ampulliform	8.5–26 × 2–4.5	Ellipsoidal	15–23 × 6–9	[1]
*P. variabile*	Cylindrical to ampulliform	12–23 × 2–4.5	Ellipsoidal	12.5–23.5 × 5.5–9	[1]

## Data Availability

The sequences from the present study were submitted to the NCBI database (https://www.ncbi.nlm.nih.gov/, accessed on 6 December 2022) and the accession numbers were listed in Table 2.

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
