# Peer review of "Taxonomy, Phylogeny, Divergence Time Estimation, and Biogeography of the Family Pseudoplagiostomataceae (Ascomycota, Diaporthales)"

_jof, 2023, doi:10.3390/jof9010082_

Round 1
Reviewer 1 Report
Please kindly see my comments and suggestions which are in the attached file.

Author Response
1. Please correct it. I accepted the suggestions. 2. Is this four or five? Five, I revised it. 3. Suggest deleting "genes". I accepted the suggestions. 4. Is this the correct full name of rpb2? Yes. 5. Please check species name. I rechecked and revised it. 6. Change Maximum Likelihood to maximum likelihood. I accepted the suggestions and revised it. 7. Italicize the name. I accepted the suggestions and revised it. 8. Is it unitunicate asci? Please describe. I accepted the suggestions and revised it. 9. growing I accepted the suggestions and revised it. 10. Delete "are"? I accepted the suggestions and revised it. 11. Delete "are" I accepted the suggestions and revised it. 12. Italicizes the name. I accepted the suggestions and revised it. 13. Change Conidiogenous cells to conidiogenous cells. I accepted the suggestions and revised it. 14. Start with capital letter? Check all I accepted the suggestions and revised it. 15. Start with capital letter? I accepted the suggestions and revised it. 16. Please correct species name. I accepted the suggestions and revised it. 17. This picture is upside down. I accepted the suggestions and revised it. 18. Start with lowercase I accepted the suggestions and revised it. 19. Delete "are" I accepted the suggestions and revised it. 20. Delete "are" I accepted the suggestions and revised it. 21. Start with lowercase I accepted the suggestions and revised it. 22. Delete "are" I accepted the suggestions and revised it. 23. reduced to conidiogenous cells I accepted the suggestions and revised it. 24. Delete "are" I accepted the suggestions and revised it. 25. Check the current name of this family. I accepted the suggestions and revised it.Reviewer 2 Report
The paper describe three new species of Pseudoplagiostoma. It also shows the common ancestor of Pseudoplagiostomataceae via molecular clock analyses and reconstructe historical biogeography by the RASP software. The paper is novelty and written in good English.
However, the fossil time point the authors used in this study is doubtful because Metacapnodium was recently proved not in Capnodiales, not even in Dothideomycetes (Sugiyama et al. 2020). The molecular clock analysis should be done again using anothrer fossil before the paper is submitted.
Ref
Sugiyama J, Nam K-O, Hosoya T (2020) Metacapnodium neesii: a new combination for a metacapnodiaceous sooty mould and its ohylogenetic position inferred from DNA sequences. Journal of Fungal Research 18 (4):246-257

Author Response
Dear sir/madam,
Thank you for your letter and comments concerning our manuscript “Taxonomy, Phylogeny, Divergence Time Estimation and Biogeography of the Family Pseudoplagiostomataceae (Ascomycota, Diaporthales)” (Submission jof-2120314).
Your comments are highly insightful and help us greatly improve the quality of our manuscript. We hope that the revisions and our responses as in the postscript would be sufficient to make our manuscript suitable for publication in Journal of Fungi.
Please check the attached documents.
Sincerely yours,
Zhaoxue Zhang

Reviewer 3 Report
Dear Authors,
In this study, three new species of the genus Pseudoplagiostoma are presented based on multi-gene region analysis. Phylonegetic analysis and morphological characters are sufficient for the new species. In addition, divergence time estimation and historical biogeography analyses for the family Pseudoplagiostomataceae increased the quality of the study.
Minor corrections and suggestions regarding the paper are marked on the text.
Best regards and a happy new year in advance,

Author Response

(The authors gave the same response as above.)

Round 2
Reviewer 2 Report
I would like to know which reference "fossil “Protocolletotrichum deccanense” set as 65 Mya for Colletotrichum" according to?
Author Response
Dear sir/madam,
Thank you for your letter and comments concerning our manuscript “Taxonomy, Phylogeny, Divergence Time Estimation and Biogeography of the Family Pseudoplagiostomataceae (Ascomycota, Diaporthales)” (Submission jof-2120314).
Kar et al. (2004) found a fossil fungus related to Colletotrichum Corda from an intertrappean bed intersected by a well at Mohgaon-Kalan village, Chhindwara District, Madhya Pradesh and named Protocolletotrichum deccanensis gen. et sp. nov. Radiometric dating of the volcanic rocks yielded ages of 67.8–61.6 Mya. So, I set as 65 Mya for Colletotrichum. For more detials, Please see the attachment.
Sincerely yours,
Zhaoxue Zhang
